# Pharmacological Modulation of the Ca^2+^/cAMP/Adenosine Signaling in Cardiac Cells as a New Cardioprotective Strategy to Reduce Severe Arrhythmias in Myocardial Infarction

**DOI:** 10.3390/ph16101473

**Published:** 2023-10-16

**Authors:** Fernando Sabia Tallo, Patricia Oliveira de Santana, Sandra Augusta Gordinho Pinto, Rildo Yamaguti Lima, Erisvaldo Amarante de Araújo, José Gustavo Padrão Tavares, Marcelo Pires-Oliveira, Lucas Antonio Duarte Nicolau, Jand Venes Rolim Medeiros, Murched Omar Taha, André Ibrahim David, Bráulio Luna-Filho, Carlos Eduardo Braga Filho, Adriano Henrique Pereira Barbosa, Célia Maria Camelo Silva, Almir Gonçalves Wanderley, Adriano Caixeta, Afonso Caricati-Neto, Francisco Sandro Menezes-Rodrigues

**Affiliations:** 1Department of Urgency and Emergency Care, Universidade Federal de São Paulo (UNIFESP), São Paulo 04024-000, SP, Brazil; 2Postgraduate Program in Cardiology, Universidade Federal de São Paulo (UNIFESP), São Paulo 04024-000, SP, Brazil; patricia.santana@unifesp.br (P.O.d.S.); sanndraagp@gmail.com (S.A.G.P.); rildoylima@gmail.com (R.Y.L.); biomedipatologia@icloud.com (E.A.d.A.); brauluna@uol.com.br (B.L.-F.); eduardo.braga@unifesp.br (C.E.B.F.); barbosa-ah@uol.com.br (A.H.P.B.); celia.maria@unifesp.br (C.M.C.S.); adriano.caixeta@unifesp.br (A.C.); 3Department of Pharmacology, Universidade Federal de São Paulo (UNIFESP), São Paulo 04023-062, SP, Brazil; padrao.tavares@hotmail.com (J.G.P.T.); caricatineto@gmail.com (A.C.-N.); 4União Metropolitana de Educação e Cultura—School of Medicine (UNIME), Lauro de Freitas 42700-000, BA, Brazil; marpoliv@umich.edu; 5Department of Biotechnology, Universidade Federal do Delta do Parnaíba (UFDPar), Parnaíba 64202-020, PI, Brazil; lucasnicolau@ufpi.edu.br (L.A.D.N.); jandvenes@ufpi.edu.br (J.V.R.M.); 6Department of Surgery, Universidade Federal de São Paulo (UNIFESP), São Paulo 04023-900, SP, Brazil; taha@uol.com.br (M.O.T.); andredavidmd@gmail.com (A.I.D.); 7Department of Pharmaceutical Sciences, Universidade Federal de São Paulo (UNIFESP), Diadema 09913-030, SP, Brazil; almir.wanderley@unifesp.br

**Keywords:** cardiac ischemia-reperfusion, cardiac arrhythmias, pharmacological cardioprotection, Ca^2+^ channels, adenosine receptors

## Abstract

Acute myocardial infarction (AMI) is the main cause of morbidity and mortality worldwide and is characterized by severe and fatal arrhythmias induced by cardiac ischemia/reperfusion (CIR). However, the molecular mechanisms involved in these arrhythmias are still little understood. To investigate the cardioprotective role of the cardiac Ca^2+^/cAMP/adenosine signaling pathway in AMI, L-type Ca^2+^ channels (LTCC) were blocked with either nifedipine (NIF) or verapamil (VER), with or without A_1_-adenosine (ADO), receptors (A_1_R), antagonist (DPCPX), or cAMP efflux blocker probenecid (PROB), and the incidence of ventricular arrhythmias (VA), atrioventricular block (AVB), and lethality (LET) induced by CIR in rats was evaluated. VA, AVB and LET incidences were evaluated by ECG analysis and compared between control (CIR group) and intravenously treated 5 min before CIR with NIF 1, 10, and 30 mg/kg and VER 1 mg/kg in the presence or absence of PROB 100 mg/kg or DPCPX 100 µg/kg. The serum levels of cardiac injury biomarkers total creatine kinase (CK) and CK-MB were quantified. Both NIF and VER treatment were able to attenuate cardiac arrhythmias caused by CIR; however, these antiarrhythmic effects were abolished by pretreatment with PROB and DPCPX. The total serum CK and CK-MB were similar in all groups. These results indicate that the pharmacological modulation of Ca^2+^/cAMP/ADO in cardiac cells by means of attenuation of Ca^2+^ influx via LTCC and the activation of A_1_R by endogenous ADO could be a promising therapeutic strategy to reduce the incidence of severe and fatal arrhythmias caused by AMI in humans.

## 1. Introduction

After 2023, cardiovascular diseases (CVD) will be responsible for over 26 million annual deaths worldwide, in both industrialized and underdeveloped nations; among these, ischemic heart disorders (IHD) and, particularly, acute myocardial infarction (AMI) are the leading cause of death and morbidity worldwide [1,2,3], estimated to affect about three million people worldwide and to significantly increase the incidence of sudden cardiac death [4,5]. AMI results in irreversible damage to the myocardium primarily caused by a lack of oxygen in cardiac cells, which may lead to impairment in diastolic and systolic function and make the patient prone to severe and fatal cardiac arrhythmias [4,5,6]. Although AMI can lead to several serious complications for cardiac function, there are still few pharmacological resources for the treatment of AMI.

The key treatment of AMI is the rapid restoration of coronary blood flow after ischemia (reperfusion) [4,5,6,7]. The earlier the treatment (less than 6 h from symptom onset), the better the prognosis. Although the main form of AMI treatment is reperfusion of the myocardium [4,5,6], this process can cause severe cardiac dysfunctions, mainly due to abrupt oxygen entry and severe ionic deregulation in cardiac cells, which in turn, may lead to lethal arrhythmias directly related to the deregulation of intracellular Ca^2+^ homeostasis in cardiac cells [6,7,8,9,10,11,12,13,14,15]. This deregulation of Ca^2+^ homeostasis results from modifications of Ca^2+^ extrusion or buffering that stimulate spontaneous Ca^2+^ release from the sarcoplasmic reticulum (SR) and that cause delayed after-depolarization activity promoted by cytosolic and mitochondrial Ca^2+^ overload during ischemia. This deregulation leads to suboptimal Ca^2+^-ATPase performance, resulting in increased cytosolic and mitochondrial Ca^2+^ concentration that collapses mitochondrial function and ATP synthesis [6,16]. These Ca^2+^-related dysfunctions induced by cardiac ischemia followed by reperfusion (CIR) frequently result in death due to a significant increase in the incidence of cardiac arrhythmias [6,16]. Most of these early-death-inducing cardiac arrhythmias consist of complex ventricular arrhythmias (VA) and atrioventricular blockades (AVB) after CIR [6,7,8,9,10,11,12,13,14,15,16,17].

Additionally, reperfusion also produces important metabolic and functional alterations in cardiac cells. It increases free radical production and Ca^2+^ influx overload through L-type Ca^2+^ channels (LTCC). This Ca^2+^ influx modulates ryanodine receptors (RyR) and important enzymes, such as adenylyl cyclase (AC), an enzyme that produces cAMP from ATP, in the T-tubules and intracellular medium [6]. The mitochondria also play a role in maintaining Ca^2+^ homeostasis during brief increases in cytosolic Ca^2+^ ([Ca^2+^]c) in cardiac cells, which is crucial in the contraction–relaxation cycle of myocardium [6,14]. Ca^2+^ concentration in the mitochondrial matrix ([Ca^2+^]m) is finely controlled by Ca^2+^ transporter proteins that are present in the mitochondrial membranes and control Ca^2+^ influx and efflux in the mitochondrial matrix [6,7]. Mitochondrial Ca^2+^ influx in cardiac cells is primarily controlled by the mitochondrial Ca^2+^ uniporter (MCU), while its efflux is mostly controlled by the mitochondrial Na^+^/Ca^2+^ exchanger (mNCX) [6,16]. As a result, the cardiac cycle and contraction–relaxation processes are significantly impacted by mitochondrial dysfunction in Ca^2+^ homeostasis in cardiac cells [6,14]. In addition to this decoupling of cardiac excitation–contraction (CECC), an important increase in free radical production during reperfusion leads to the oxidation of structural proteins, proteins involved in the respiratory chain, pyridine nucleotides, changes in permeability of the internal mitochondrial membrane, decoupling of oxidative phosphorylation, and a decrease in mitochondrial ATP production [6,14].

In addition to its role in CECC, Ca^2+^ modulates 3′5′-cyclic adenosine monophosphate (cAMP) production with isoforms 5 and 6 of adenylyl cyclase (AC), and the pharmacological blockade of Ca^2+^ influx via LTCC produces an increase in production and efflux of intracellular cAMP in cardiac cells [6,18,19,20]. In the extracellular medium, cAMP is transformed into adenosine (ADO), which then stimulates membrane A_1_-type ADO receptors (A_1_R) to finely regulate cardiac cell function [6]. It is well known that stimulation of cardiac A_1_R by ADO is a common and effective strategy used to attenuate cardiac arrhythmias in various clinical situations, and especially in cardiac surgery [4,6]. Thus, we have proposed that pharmacological modulation of Ca^2+^/cAMP/ADO signaling in cardiac cells could be a promising strategy in the treatment of AMI and other IHD in humans.

Based on the above proposal, in the present work, we investigated the effects of pharmacological modulation Ca^2+^/cAMP/ADO signaling in cardiac cells on the incidence of severe and fatal arrhythmias related to AMI. Thus, using an animal model of AMI, the effects of the blockade of LTCC-mediated Ca^2+^ influx with nifedipine (NIF) or verapamil (VER), in the presence or absence of blocker of transporter-mediated cAMP efflux probenecid (PROB) or A_1_R-selective antagonist 8-cyclopentyl-1,3-dipropylxanthine (DPCPX), on the incidence of arrhythmias (VA and AVB) and lethality (LET) induced by CIR were studied (Figure 1 and Figure 2). In addition, serum concentrations of cardiac injury biomarkers total creatine kinase (CK) and CK-MB were quantified.

## 2. Results

### 2.1. Incidence of VA, AVB and LET Induced by CIR

No arrhythmias were detected during the stabilization periods of any animal (15 min). During CIR, VA and AVB were detected and measured in different experimental groups. After CIR, the incidence of VA, AVB and LET were 90%, 80% and 70%, respectively (Figure 2).

### 2.2. Effects of the NIF and VER on the Incidence of VA, AVB and LET Induced by CIR

Figure 2 shows that incidences of AVB and LET induced by CIR were significantly reduced by treatment with NIF (1, 10 and 30 mg/kg, IV) and VER (1 mg/kg, IV). VA incidence was reduced from 90% to 30% in NIF10 + CIR and 30% in NIF30 + CIR groups, compared to CIR group. AVB incidence was reduced from 80% to 30% in NIF1 + CIR, 20% in NIF10 + CIR, and 20% in NIF30 + CIR groups, compared to CIR group. LET incidence was reduced from 70% to 30% in NIF1 + CIR, 10% in NIF10 + CIR, and 20% in NIF30 + CIR, compared to CIR group. In addition, treatment with VER was also able to reduce the incidences of VA (90% to 20%), AVB (90% to 20%), and LET (90% to 20%) induced by CIR. These results confirm previous studies [6,7,11] that demonstrated that the blockade of Ca^2+^ influx via LTCC in cardiac cells before CIR attenuates cardiac collapse and reduces the incidence of severe and fatal arrhythmias induced by CIR.

### 2.3. Effects of Pretreatment with PROB or DPCPX before Administration of NIF or VER on the Incidence of VA, AVB and LET Induced by CIR

To investigate whether Ca^2+^/cAMP/ADO signaling in cardiac cells is involved in the cardioprotective effect of NIF and VER (Figure 2), we pretreated rats subjected to CIR with DPCPX (100 µg/kg, IV) or PROB (100 mg/kg, IV), as well as NIF (10 mg/kg, IV) and VER (1 mg/kg, IV). Figure 3 shows that the reduction of VA, AVB and LET incidence in the PROB + NIF + CIR, PROB + VER + CIR, DPCPX + NIF + CIR, and DPCPX + VER + CIR groups was not statistically different from the CIR group, indicating that pretreatment with DPCPX and PROB completely abolished the cardioprotective effects of NIF and VER. These results indicate that an increase in extracellular levels of ADO due to cAMP transport to extracellular environments combined with an increase in activation of A_1_R receptors in cardiac cells directly participates in the cardioprotective response stimulated by NIF and VER in rats subjected to CIR.

### 2.4. Effects of Pretreatment with DPCPX before Administration of NIF and VER on Biochemical Markers of Cardiac Injury

Figure 4A shows that the serum concentration of biomarkers of cardiac lesion, total CK and CK-MB, were not statistically different in CIR (5487 ± 449 mg/dL, *n* = 3), NIF1 + CIR (5395 ± 876 mg/dL, *n* = 5), NIF10 + CIR (5344 ± 193 mg/dL, *n* = 5), NIF30 + CIR (5018 ± 508 mg/dL, *n* = 5), DPCPX + NIF + CIR (4864 ± 445 mg/dL, *n* = 5), VER + CIR (4437 ± 771 mg/dL, *n* = 5), and DPCPX + VER + CIR (4802 ± 254 mg/dL, *n* = 5) groups. Similarly, Figure 4B shows that serum CK-MB concentrations were also not statistically different in CIR (2225 ± 290 mg/dL, *n* = 3), NIF1 + CIR (2087 ± 61 mg/dL, *n* = 5), NIF10 + CIR (2054 ± 106.3 mg/dL, *n* = 5), NIF30 + CIR (2112 ± 102 mg/dL, *n* = 5), DPCPX + NIF + CIR (1954 ± 161 mg/dL, *n* = 5), VER + CIR (2905 ± 656 mg/dL, *n* = 5), and DPCPX + VER + CIR (2701 ± 350 mg/dL, *n* = 5) groups. Thus, NIF, DPCPX, and VER seem to modulate cardiac electric activity to attenuate arrhythmias and LET post-CIR, with no impact on the extent or severity of ischemic cell lesions.

## 3. Discussion

In the present work, we showed that pharmacological modulation of Ca^2+^/cAMP/ADO signaling in cardiac cells by means of attenuation of Ca^2+^ influx via LTCC combined with an increase in the activation of A_1_R by ADO generated from extracellular cAMP reduced the incidence of severe and fatal arrhythmias induced by CIR (see Figure 2 and Figure 3). Similar pharmacological approaches using LTCC blockers and stimulation of cardiac A_1_R have been used for cardiac arrhythmias in various clinical situations, especially in cardiac surgery [4,6]. Thus, pharmacological modulation of Ca^2+^/cAMP/ADO signaling in cardiac cells could be a promising therapeutic strategy to also reduce the incidence of severe and fatal arrhythmias caused by AMI in humans.

The dynamic equilibrium between the concentration of Ca^2+^ into cytosol, sarcoplasmic reticulum and mitochondria is crucial to finely control cardiac excitation–contraction coupling (CECC) [6,8,9]. Thus, deregulation of cellular Ca^2+^ homeostasis causes decoupling of CECC, increasing the incidence of cardiac arrhythmias [6]. ATP deficit during ischemia inhibits ATP-dependent ionic transporters, like Na^+^/K^+^-ATPase, Ca^2+^-ATPase plasmalemmal (PMCA), and sarco-endoplasmic reticulum Ca^2+^-ATPase (SERCA), leading to the accumulation of Na^+^ and Ca^2+^ in the cytosol [6,14] and cytosolic Ca^2+^ overload [6,14]. This process induces an increase in mitochondrial Ca^2+^ influx, which further reduces ATP production and, consequently, collapses the cardiac function [6]. Cytosolic and mitochondrial Ca^2+^ overload severely compromises CECC, favoring the development of severe and fatal arrhythmias [6,8,9,20]. Thus, drugs that reduce Ca^2+^ influx through L-type Ca^2+^ channels (LTCC) in cardiac cells significantly reduced the incidence of severe and fatal arrhythmias induced by CIR, strengthening the idea that the attenuation of cytosolic and mitochondrial Ca^2+^ overload reduces cardiac collapse caused by CIR [6].

In addition to its role in CECC, LTCC-mediated Ca^2+^ influx in cardiac cells modulates cAMP production by AC isoforms 5 (AC5) and 6 (AC6) [6], the activation of adrenergic receptors in the heart, and pharmacological block of Ca^2+^ via LTCC increases production and efflux of intracellular cAMP [6,21]. In the extracellular medium, cAMP is transformed into ADO that can stimulate A_1_R located in the plasma membrane of cardiac cells to finely regulate cardiac function [6,22]. Biochemical analyses of membrane preparations in overexpression systems have been used to establish the paradigm for Ca^2+^-mediated inhibition of AC5 and AC6 in the submicromolar range [6]. In fact, the crystal structure of an AC5-catalytic domain-containing high affinity Ca^2+^-pyrophosphate (PPi) complex was just recently published [20]. Although there are many papers reporting Ca^2+^-mediated modulation of AC6 activity in an endogenous setting, whole cell overexpression experiments provide most of the evidence for Ca^2+^ inhibition of AC6 [6,19]. Although less thorough, the evidence for Ca^2+^-mediated inhibition of the extremely comparable AC5 is consistent in its assertion. As a result, there is strong support for the idea that Ca^2+^ inhibits both AC5 and AC6 and that this inhibition occurs both in vitro and in vivo [6,19].

The physiological effects of knocking down AC5 and AC6 have been seen in several investigations, although none of these can be clearly linked to the enzymes’ Ca^2+^-susceptibility to inhibition by Ca^2+^. Mice lacking the AC5 gene have impaired pain perception, diminished motor activity, and altered heart function [6,19,20]. The preponderance of AC5 and AC6 in cardiac tissue has been hypothesized to have a substantial role in the rhythmicity of sympathetic regulation of inotropy [20]. AC5 null animals exhibit a lower left ventricular ejection fraction, attenuated baroreflexes, and a lack of acetylcholine-mediated Gi inhibition of AC activity [20]. They also have reduced Ca^2+^-mediated inhibition of cAMP. Reduced left ventricular function is shown in AC6 knockout mice, as well as a diminished Ca^2+^-mediated suppression of cAMP [20].

Additionally, cytosolic Ca^2+^ also regulates several intracellular second messengers and various cellular responses [6,22]. For instance, increased Ca^2+^ influx through LTCC modulates the β_1_-adrenoceptors (β_1_AR)-mediated excitatory response of cardiac cells, due to the Ca^2+^-induced inhibition of AC activity [6,21,22]. Thus, cytosolic cAMP increases due to cardiac β_1_AR stimulation are even higher when Ca^2+^ influx is decreased, such as by the action of LTCC blockers [6,21,22], thus possibly leading to increased cAMP efflux and ADO production following sympathetic stimulation [6,21,22].

In stress conditions, such as hypoxia or ischemia, increased extracellular ADO levels are responsible for cardioprotective effects, which involve, at least in part, the activation of Gi-coupled A_1_R [23,24,25]. The activation of A_1_R and A_3_R has been shown to decrease cardiac infarction lesion size, as well as to consistently improve functional recovery in isolated hearts [22,23,24,25,26,27,28]. A_1_R mediates the direct negative chronotropic and dromotropic actions of ADO, as well as indirect anti-β_1_AR actions [21,29,30,31,32]. It is significant to note that pharmacological stimulation of cardiac A_1_R lowers cardiac cell excitability [33,34,35,36], perhaps reducing the likelihood of fatal AVB. There is substantial evidence that the activation of all four AR (A_1_, A_2A_, A_2B_ and A_3_) is importantly involved in cardioprotective response in different pathological conditions, including the CIR [33,34].

In adult male rats subjected to in vivo regional myocardial ischemia (25 min) and reperfusion (120 min), treatment with the A_1_R-selective agonist 2-chloro-N6-cyclopentyladenosine (CCPA) (10 µg/kg) or the nonselective AR agonist 5′-N-Ethylcarboxamidoadenosine (NECA) (10 µg/kg) reduced myocardial infarction size by 50% and 35%, respectively [28]. These cardioprotective effects were blocked by pretreatment with selective antagonists of A_1_R (DPCPX, 100 µg/kg) or A_2a_R (ZM241385, 1.5 mg/kg) [28]. In a cardiac H9c2(2-1) cell ischemia model, AR agonists, such as N6-cyclopentyladenosine (CPA) and (N(6)-(3-iodobenzyl)-adenosine-5′-N- methylcarboxamide (IB-MECA), reduced the proportion of nonviable cells [34]. However, these cardioprotective effects of CPA were decreased in the presence of ADO deaminase, which reduces the endogenous levels of ADO [34]. In addition, these cardioprotective effects mediated by AR were also attenuated by DPCPX, ZM241385 or A_2b_R-selective antagonist MRS1191 [34]. Despite the likely cardioprotective role of exogenous or endogenous ADO in the prevention of cardiomyocyte necrosis, in this work, we observed a significant antiarrhythmic effect produced by endogenous ADO and LTCC blockers NIF and VER (see Figure 2 and Figure 3), independent of any effect on biochemical markers of cardiomyocyte lesion (see Figure 4).

The results obtained in this study (see Figure 2 and Figure 3) also demonstrate that there is a positive correlation between Ca^2+^ influx via LTCC and activity of the purinergic pathway through A_1_R activation in cardiac cells. This cross-communication between Ca^2+^ influx and the purinergic signaling mediated by A_1_R is importantly involved in the regulation of the electrophysiology and contractile activity of cardiac cells, attenuating the severe and fatal arrhythmias induced by CIR. It was shown that the positive chronotropic response induced by the activation of cardiac β_1_AR is attenuated by an increase in extracellular levels of ADO produced by the enzymatic degradation of ATP released from intracardiac sympathetic neurons combined with the transport of cAMP to the extracellular medium from cardiac cells during stimulation [6,21,22]. According to several lines of evidence, the adrenergic–purinergic communication that is critical for controlling cardiac chronotropism also plays a significant role in cardioprotective responses under various pathological circumstances [6,22,35]. However, like other xanthine derivatives, DPCPX also functions as a phosphodiesterase (PDE) inhibitor and is virtually as powerful as rolipram at inhibiting PDE [6,22,35]. DPCPX exhibits a high selectivity for A_1_R over other AR subtypes [35,36]. Figure 3 shows that DPCPX inhibited the cardioprotective effects of NIF and VER, indicating that A_1_R is involved in these effects (see Figure 3).

We have proposed that this pharmacological modulation of Ca^2+^/cAMP/ADO signaling in cardiac cells by means the attenuation of Ca^2+^ influx via LTCC combined with an increase in the activation of A_1_R by ADO generated by the increment of extracellular transport of cAMP may be effective to prevent sudden mortality in individuals with AMI due to severe arrhythmias brought on by cardiac collapse. Bringing together all the results obtained in the present study and the existing data in the literature, we built a theoretical model of cardioprotective response stimulated by pharmacological modulation of the Ca^2+^/cAMP/ADO signaling in cardiac cells (see Figure 5).

## 4. Materials and Methods

### 4.1. Animals

Male Wistar rats (14- to 16-week-old) weighing between 290 and 320 g, were kept at 21 ± 2 °C with a 12:12 h light/dark cycle and were given food and water ad libitum. All experimental protocols used in this study were approved by the Ethics Committee of the Escola Paulista de Medicina—Universidade Federal de São Paulo (UNIFESP #1130/11 and #0065/12).

### 4.2. Cardiac Ischemia and Reperfusion (CIR) Induction

To produce an animal model of AMI, rats were subjected to surgical procedures in accordance with the approach previously published by our laboratory [7,8,9]. Initially, the rats were anesthetized with ketamine (75 mg/kg, intraperitoneally) and xylazine (8 mg/kg, intraperitoneally). After anesthesia, rats were intubated using a Jelco 14G catheter (New York, NY, USA), and mechanically ventilated using an Insight model EFF 312 mechanical ventilator (Insight Equipamentos Cientificos, Ribeirão Preto, SP, Brazil) [7,8,9]. A thoracotomy was carried out to insert a vascular tourniquet (4/0 braided silk suture linked to a 10 mm micropoint reverse cutting needle, Ethicon K-890H, Cincinnati, OH, USA) around the left anterior descending coronary artery to induce ischemia after the animal had been stabilized for 15 min. The tourniquet was removed after 10 min of myocardial ischemia to allow 75 min of coronary recirculation (cardiac reperfusion) [7,8,9].

### 4.3. Assessment of Cardiac Activity during CIR

All animals underwent ECG analysis to evaluate cardiac activity during CIR, in accordance to previously described methodology [7,8,9,10,11,12]. This ECG analysis was performed to evaluate the effects of NIF and VER, in the presence or absence of blocker ABC transporter-mediated cAMP efflux PROB or A_1_R-selective antagonist DPCPX, on the incidence of arrhythmias (VA and AVB) and lethality (LET) induced by CIR. Initially, ECG was recorded for 15 min before ischemia protocol (stabilization period) and during ischemia (10 min) and reperfusion (75 min) protocol [7,8,9,10,11,12]. A biopotential amplifier was used to record the ECG using needle electrodes inserted subcutaneously on the limbs. ECG changes (increase in R wave and ST segment) brought on by CIR were utilized to confirm that the coronary artery had successfully been blocked by surgery [7,8,9,10]. A heated operating table and the proper heating lamps were used to keep body temperature at 37.5 °C, and a rectal thermometer was regularly used to check the temperature [7,8,9,10]. ECG data were captured using the AqDados 7.02 collection equipment from Lynx Tecnologia Ltda. (São Paulo, Brazil) and examined using AqDAnalysis 7 software. We assessed heart rates and the incidence of VA, AVB, and LET brought on by CIR using this program. All three conditions were regarded as VA: ventricular fibrillation, *torsades de pointes*, and ventricular tachycardia [8,9].

### 4.4. Biochemical Assessment of Biomarkers of Cardiac Lesion

Blood samples (3–4 mL) were taken from the abdominal aorta and placed in siliconized tubes to determine the serum levels of biomarkers of cardiac lesion, total creatine kinase (CK), and creatine kinase–MB fraction (CK-MB). These samples were taken from rats that survived the entire 75 min CIR protocol. Centrifugation of blood samples (2500 rpm for 40 min at 5 °C) was performed and the supernatant was removed and kept at −20 °C for enzymatic analysis. A commercial kit from Vida Biotecnologia, Belo Horizonte, Brazil, was used to perform a kinetic UV test, measuring at 340 nm the enzymatic activity of CK and CK-MB in serum [8].

### 4.5. Pharmacological Treatments

To evaluate the effects of nifedipine (NIF) (1 mg/kg, 10 mg/kg, and 30 mg/kg, IV; Sigma-Aldrich, St. Louis, MO, USA) and verapamil (VER) (1 mg/kg, IV; Sigma-Aldrich, St. Louis, MO, USA) on the incidence of VA, AVB and LET caused by CIR, rats were treated with this NIF or VER alone or combined with drugs that block ABC transporter-mediated cAMP efflux from cardiac cells (probenecid, PROB; Sigma-Aldrich, St. Louis, MO, USA) or A_1_R antagonist (8-cyclopentyl-1,3-dipropylxanthine, DPCPX; Sigma-Aldrich, St. Louis, MO, USA). All drugs were intravenously (IV) administered before CIR. We previously showed that LET in control animals treated with 0.9% saline solution (SS) varied from 70–80% [8,9]. In the present work, the animals used were divided into the following experimental groups:CIR group (*n* = 40): animals were treated with 0.9% saline solution (IV) one minute before they were subjected to a surgical procedure to induce cardiac ischemia (10 min), followed by coronary reperfusion (75 min), and subsequent ECG monitoring (100 min) for determination of VA, AVB and LET incidence;PROB + CIR group (*n* = 20): animals were treated with ABC transporter-mediated cAMP efflux blocker probenecid (PROB, 100 mg/kg, IV), five minutes before they were subjected to a surgical procedure to induce cardiac ischemia (10 min), followed by coronary reperfusion (75 min), and subsequent ECG monitoring (100 min) for determination of VA, AVB and LET incidence;NIF1 + CIR group (*n* = 20): animals were treated with NIF (1 mg/kg, IV) one minute before they were subjected to a surgical procedure to induce cardiac ischemia (10 min), followed by coronary reperfusion (75 min), and subsequent ECG monitoring (100 min) for determination of VA, AVB and LET incidence;NIF10 + CIR group (*n* = 20): animals were treated with NIF (10 mg/kg, IV) one minute before they were subjected to a surgical procedure to induce cardiac ischemia (10 min), followed by coronary reperfusion (75 min), and subsequent ECG monitoring (100 min) for determination of VA, AVB and LET incidence;NIF30 + CIR group (*n* = 20): animals were treated with NIF (30 mg/kg, IV) one minute before they were subjected to a surgical procedure to induce cardiac ischemia (10 min), followed by coronary reperfusion (75 min), and subsequent ECG monitoring (100 min) for determination of VA, AVB and LET incidence;PROB + NIF + CIR group (*n* = 20): animals were pretreated with ABC transporter-mediated cAMP efflux blocker probenecid (PROB; 100 mg/kg, IV), then NIF (10 mg/kg, IV) 5 min later; then, 1 min later, cardiac ischemia was induced (10 min), followed by cardiac reperfusion, and subsequent ECG monitoring for 100 min for determination of the VA, AVB and LET incidence;DPCPX + NIF + CIR group (*n* = 20): animals were pretreated with A_1_R antagonist 8-Cyclopentyl-1,3-dipropylxanthine (DPCPX; 100 µg/kg, IV), then NIF (10 mg/kg, IV) 5 min later; then, 1 min later, cardiac ischemia was induced (10 min), followed by cardiac reperfusion, and subsequent ECG monitoring for 100 min for determination of the VA, AVB and LET incidence;VER1 + CIR group (*n* = 20): animals were treated with verapamil (VER, 1 mg/kg, IV) for 1 min before induction of cardiac ischemia (10 min), followed by coronary reperfusion (75 min), and subsequent ECG monitoring (100 min) for determination of the VA, AVB and LET incidence.PROB + VER + CIR group (*n* = 20): animals were pretreated with ABC transporter-mediated cAMP efflux blocker probenecid (PROB; 100 mg/kg, IV), then VER (1 mg/kg, IV) 5 min later; then, after 1 min, cardiac ischemia was induced (10 min), followed by cardiac reperfusion, and subsequent ECG monitoring for 100 min for determination of the VA, AVB and LET incidence;DPCPX + VER + CIR group (*n* = 20): animals were pretreated with A_1_R antagonist 8-Cyclopentyl-1,3-dipropylxanthine (DPCPX; 100 µg/kg, IV), then VER (1 mg/kg, IV) 5 min later; then, after 1 min, cardiac ischemia was induced (10 min), followed by cardiac reperfusion, and subsequent ECG monitoring for 100 min for determination of the VA, AVB and LET incidence.

### 4.6. Data Analysis

Data corresponding to VA, AVB, and LET incidences were expressed as percentages and statistically compared using Fisher’s exact test with the Prism 5.0 software (GraphPad, San Diego, CA, USA) [7,8,9,10,11,12]. Data corresponding to the serum concentration (mg/dL) of biomarkers of cardiac lesion (total CK and CK-MB) were expressed as the mean ± the standard error of the mean (SEM) and statistically analyzed with an analysis of variance test using Prism [8]. The results were considered statistically significant when *p* < 0.05.

## 5. Conclusions

The results obtained in the present study indicate that pharmacological modulation of Ca^2+^/cAMP/ADO signaling in cardiac cells by means of the attenuation of Ca^2+^ influx via LTCC and the activation of A_1_R by endogenous ADO could be a promising therapeutic strategy to reduce the incidence of severe and fatal arrhythmias caused by AMI in humans.

## Figures and Tables

**Figure 1 pharmaceuticals-16-01473-f001:**
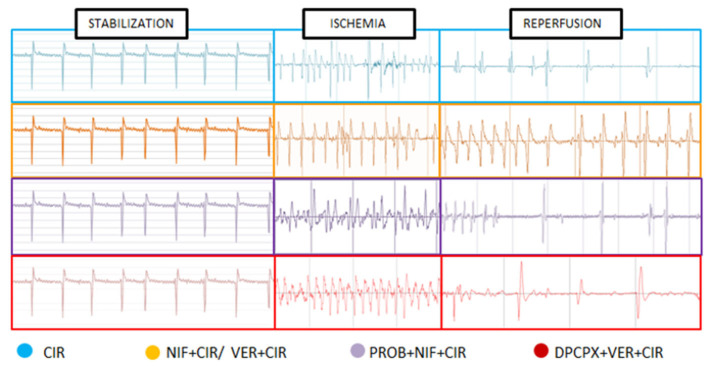
(Blue) ECG recording of control rats submitted to cardiac ischemia and reperfusion (CIR) protocol; (Yellow) ECG recording of rats treated with nifedipine or/and verapamil before of CIR; (Purple) ECG recording of rats treated with probenecid and nifedipine before of CIR; (Red) ECG recording of rats treated with probenecid and 8-cyclopentyl-1,3-dipropylxanthine (DPCPX) before of CIR.

**Figure 2 pharmaceuticals-16-01473-f002:**
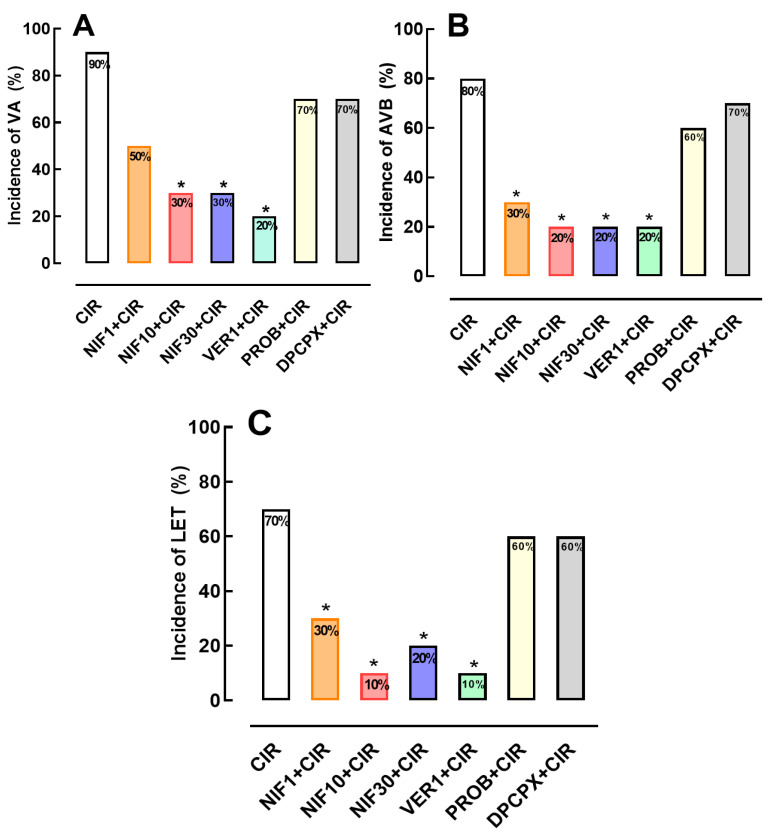
Incidence of (**A**) ventricular arrhythmias (AV), (**B**) atrioventricular block (AVB), and (**C**) lethality (LET) in the CIR, NIF1 + CIR, NIF10 + CIR, NIF30 + CIR, VER1 + CIR, PROB + CIR, and DPCPX + CIR groups. The incidences of VA, AVB and LET were significantly reduced in all groups of animals treated with NIF (1, 10 and 30 mg/kg or VER 1 mg/kg) pre-CIR, when compared with vehicle-treated CIR animals. Groups were compared using Fisher’s exact test (* *p* < 0.05).

**Figure 3 pharmaceuticals-16-01473-f003:**
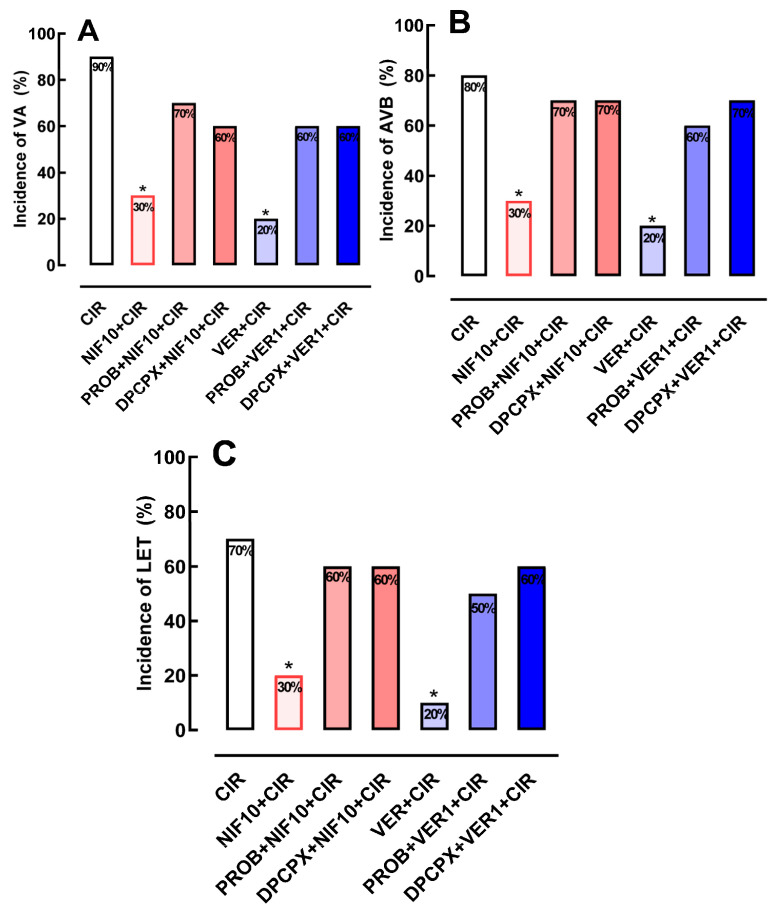
Incidence of (**A**) ventricular arrhythmias (AV), (**B**) atrioventricular block (AVB) and (**C**) lethality (LET) in the CIR, NIF10 + CIR, PROB + NIF10 + CIR, DPCPX + NIF10 + CIR, VER1 + CIR, PROB + VER1 + CIR, and DPCPX + VER1 + CIR groups. Pretreatment with either PROB or DPCPX 4 min before administration of NIF or VER was able to abolish the cardioprotective effects of NIF and VER in rats subjected to CIR. Groups were compared using Fisher’s exact test (* *p* < 0.05).

**Figure 4 pharmaceuticals-16-01473-f004:**
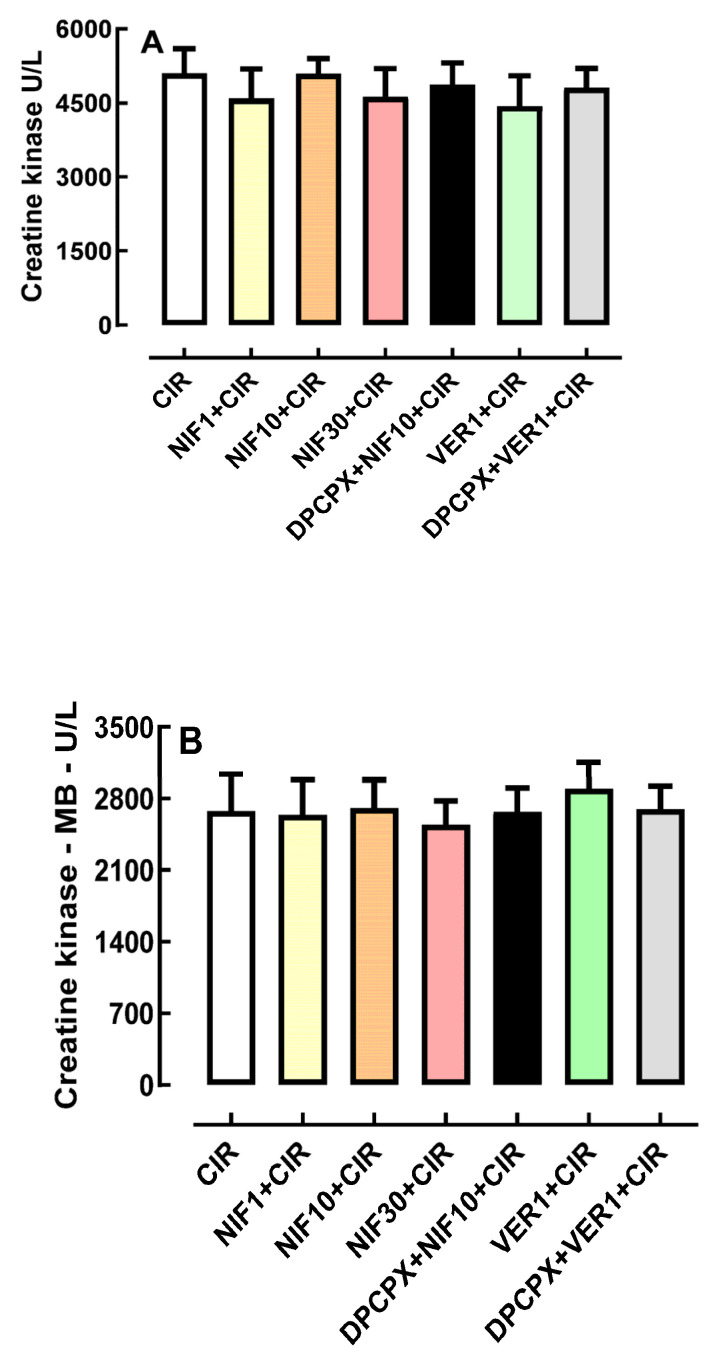
Histograms representing the serum concentrations of total CK (**A**) and CK-MB (**B**) in the CIR, NIF + CIR, NIF10 + CIR, NIF30 + CIR, DPCPX + NIF10 + CIR, VER 1 + CIR, and DPCPX + VER1 + CIR groups. Results were expressed as mean ± standard error of mean, and one-way analysis of variance (ANOVA) was applied, followed by Tukey’s post-test. There was no statistical difference between the different groups.

**Figure 5 pharmaceuticals-16-01473-f005:**
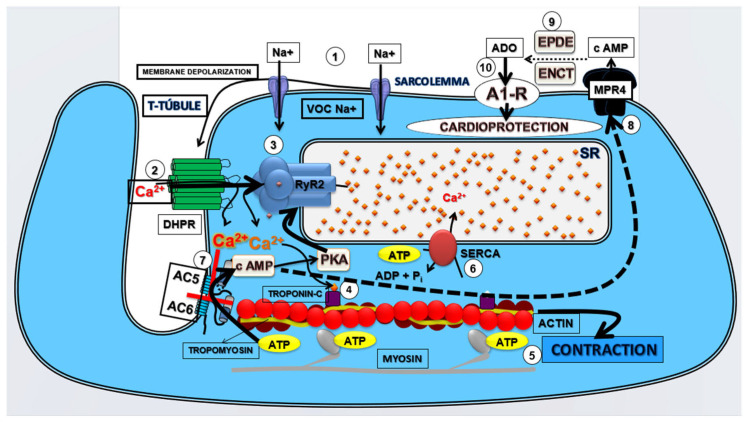
Theoretical model of cardioprotective response stimulated by pharmacological modulation of the Ca^2+^/cAMP/ADO signaling in cardiac cells. The membrane depolarization of cardiac cells generates the (1) Na^+^ influx via voltage-operated Na^+^ channels (VOC Na^+^) that induces (2) opening of the L-type Ca^2+^ channels (LTCC) containing dihydropyridine receptors (DHPR) located in the T-tubules and depolarization of the sarcolemma that lead to Ca^2+^ influx that (3) stimulates the release of Ca^2+^ from sarcoplasmic reticulum (SR) mediated by activation by Ca^2+^ of the R_2_-type ryanodine receptors (RYR_2_); this Ca^2+^ released from SR (4) binds to troponin C and (5) activates the contractile machinery generating contraction of cardiac cells. At the same time, (6) the Ca^2+^ concentration in the cytosol is restored by Ca^2+^ sequestration by SR via SERCA; Ca^2+^ influx via LTCC also (7) inhibits the isoforms 5 and 6 of adenylyl cyclase (AC), producing reduction in the intracellular production of cAMP and phosphorylation of RYR_2_ by cAMP-dependent kinases (PKA), and then reducing cardiac contraction frequency and strength; in addition (8), efflux of cAMP through membrane transporters MRP4 and (9) extracellular degradation of cAMP to adenosine (ADO) by ectonucleotides (ENCT) and ectophosphodiesterases (EPDE) increases extracellular ADO levels that lead to (10) activation of A_1_R, culminating in the cardioprotection resulting from prevention of CECC collapse and consequent reduction of incidence of severe and fatal cardiac arrhythmias induced by CIR.

## Data Availability

Data is contained within the article.

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
