# Peer review of "Pharmacological Modulation of the Ca2+/cAMP/Adenosine Signaling in Cardiac Cells as a New Cardioprotective Strategy to Reduce Severe Arrhythmias in Myocardial Infarction"

_pharmaceuticals, 2023, doi:10.3390/ph16101473_

Round 1

Reviewer 1 Report

The manuscript is devoted to a current topic in biomedicine - cardiovascular diseases and cardioprotection. The relevance of the information obtained is beyond doubt, but the manuscript has a number of shortcomings. Some of them are noted below. In addition, the presentation of the results as a whole seems to the reviewer to be unsuccessful. I believe that the article can be accepted for publication after revision

1. The graphs in some figures do not show statistics (Figures 1 and 2). Without this, it is impossible to draw a conclusion about the statistical reliability of a look at the Graphs.

2. The methods section does not sufficiently and clearly describe methods for determining biomarkers in the blood and especially techniques for assessing cardiac damage (Figures 1 and 2). Perhaps the authors should provide a diagram of the criteria for evaluating each incident and (or) provide an example of an ECG graph

3. The scheme in Figure 4 is completely unreadable

Author Response

Dear Reviewer,

These are our answers to the questions raised.

The manuscript is devoted to a current topic in biomedicine - cardiovascular diseases and cardioprotection. The relevance of the information obtained is beyond doubt, but the manuscript has a number of shortcomings. Some of them are noted below. In addition, the presentation of the results as a whole seems to the reviewer to be unsuccessful. I believe that the article can be accepted for publication after revision.

  1. The graphs in some figures do not show statistics (Figures 1 and 2). Without this, it is impossible to draw a conclusion about the statistical reliability of a look at the Graphs.

Answer: Firstly, thank you very much for your considerations. Furthermore, we modified the article with the aim of solving the problem highlighted.

  1. The methods section does not sufficiently and clearly describe methods for determining biomarkers in the blood and especially techniques for assessing cardiac damage (Figures 1 and 2). Perhaps the authors should provide a diagram of the criteria for evaluating each incident and (or) provide an example of an ECG graph.

Answer: Firstly, thank you very much for your considerations. Furthermore, we modified the article with the aim of solving the problem highlighted.

  1. The scheme in Figure 4 is completely unreadable

Answer: Firstly, thank you very much for your considerations. Furthermore, we believe that the problem that occurred was the deformatting of the figure, which certainly made it difficult to understand the information contained in the figure, which is why we modified the figure with the aim of solving the problem highlighted.

Reviewer 2 Report

The paper investigates the cardioprotective role of the cardiac Ca2+/cAMP/adenosine signaling pathway in acute myocardial infarction (AMI) and its potential to reduce the incidence of severe and fatal arrhythmias. The study uses L-type Ca2+ channel blockers (Nifedipine and Verapamil) in combination with A1-adenosine (ADO) receptor antagonists (DPCPX) or transporter-mediated cAMP efflux blocker (Probenecid) to evaluate their effects on ventricular arrhythmias (VA), atrioventricular block (AVB), lethality (LET), and cardiac injury biomarkers in rats subjected to cardiac ischemia/reperfusion (CIR). I only have several minor comments and suggestions:

  1. For Figures 1 and 2, the author should also present the data as Mean ± SD, as the data shown in Figure 3.

  2. The author should provide a high-resolution image of Figure 4.  

  3. The paper should provide more detailed discussions in these areas. It would benefit from discussing the broader clinical implications of the study's results and briefly addressing its limitations.

  4. A brief mention of the study's limitations, such as the choice of animal model or potential confounding factors, should be included to provide a more balanced view of the research. And, it would be beneficial to state this conclusion more explicitly, while also acknowledging the need for further research and clinical validation.

Author Response

Dear Reviewer,

These are our answers to the questions raised.

The paper investigates the cardioprotective role of the cardiac Ca2+/cAMP/adenosine signaling pathway in acute myocardial infarction (AMI) and its potential to reduce the incidence of severe and fatal arrhythmias. The study uses L-type Ca2+ channel blockers (Nifedipine and Verapamil) in combination with A1-adenosine (ADO) receptor antagonists (DPCPX) or transporter-mediated cAMP efflux blocker (Probenecid) to evaluate their effects on ventricular arrhythmias (VA), atrioventricular block (AVB), lethality (LET), and cardiac injury biomarkers in rats subjected to cardiac ischemia/reperfusion (CIR). I only have several minor comments and suggestions:

  1. For Figures 1 and 2, the author should also present the data as Mean ± SD, as the data shown in Figure 3.

Answer: Firstly, thank you very much for your considerations. Furthermore, Use Fisher's exact test to analyze a 2x2 contingency table and test whether the row variable and the column variable are independent, which is why we do not put Mean ± SD.

  1. The author should provide a high-resolution image of Figure 4.  

Answer: Firstly, thank you very much for your considerations. Furthermore, we believe that the problem that occurred was the deformatting of the figure, which certainly made it difficult to understand the information contained in the figure, which is why we modified the figure with the aim of solving the problem highlighted.

  1. The paper should provide more detailed discussions in these areas. It would benefit from discussing the broader clinical implications of the study's results and briefly addressing its limitations.

Answer: Firstly, thank you very much for your considerations. Furthermore, we modified the article      with the aim of solving the problem highlighted.

  1. A brief mention of the study's limitations, such as the choice of animal model or potential confounding factors, should be included to provide a more balanced view of the research. And, it would be beneficial to state this conclusion more explicitly, while also acknowledging the need for further research and clinical validation.

Answer: Firstly, thank you very much for your considerations. Furthermore, we modified the article      with the aim of solving the problem highlighted.
